# Utility of Functional Lumen Imaging Probe in Long-Term Follow-Up of Children with Esophageal Atresia: A Single-Center Retrospective Study

**DOI:** 10.3390/children9101426

**Published:** 2022-09-20

**Authors:** Francesca Destro, Sara Costanzo, Eleonora Durante, Maria Sole Carcassola, Milena Meroni, Marco Brunero, Angela Riccio, Valeria Calcaterra, Gloria Pelizzo

**Affiliations:** 1Department of Pediatric Surgery, “Vittore Buzzi” Children’s Hospital, 20154 Milan, Italy; 2Department of Pediatrics, “Vittore Buzzi” Children’s Hospital, 20154 Milan, Italy; 3Department of Internal Medicine and Therapeutics, University of Pavia, 27100 Pavia, Italy; 4Department of Biomedical and Clinical Science, University of Milano, 20157 Milan, Italy

**Keywords:** esophageal function, esophageal atresia, EndoFlip™, pediatric surgery

## Abstract

Long term follow-up of patients with esophageal atresia (EA) may be hampered by esophageal dysmotility, which affects quality of life and might lead to late complications. The endoluminal functional lumen imaging probe (EndoFlip™ Medtronic, Crospon Inc.) is an innovative diagnostic tool that assesses esophageal distensibility. Our aim was to report the use of EndoFlip™ in an EA follow-up, in order to describe distensibility patterns and to determine its possible role for functional evaluation of patients. We retrospectively collected data of EA patients, with a minimum follow-up of 9 years, who required endoscopic evaluation and underwent EndoFlip™. An adaptation of the Medtronic EF-322 protocol was applied and distensibility data were compared to those reported by Pandolfino et al. Nine patients (median age 13 years) were included in the study. The median minimum distensibility was 2.58 mm^2^/mmHg. Signs of peristalsis were observed in three patients. In one case, the esophagogastric junction (EGJ) after Toupet fundoplication showed low distensibility. EGJ distensibility values of 2.58 mm^2^/mmHg (median) confirmed both good esophagogastric continence and compliance. Esophagitis and absent peristalsis were found in one patient together with partial stenosis of the fundoplication, confirming the importance of surgical adaptation. Esophageal body distensibility was higher than that of the EGJ. Considering the presence of symptoms, the EndoFlip™ results seem to correlate better with the clinical picture. EndoFlip™ use was safe and feasible in children. It allowed for the measurement of esophageal distensibility and diameter and the acquisition of indirect information on motility with clinical implications. The routine use of EndoFlip™ could be part of EA follow-up, although considerable research is needed to correlate Endoflip™ system measurements to EA patient outcomes.

## 1. Introduction

Esophageal atresia (EA) is a rare congenital disease that has now achieved good survival rates (>90%, according to recent papers) [1,2,3]. For this reason, while in the past the focus was mainly on improving the surgical outcome and life expectancy in the neonatal period, the interest recently has shifted to the study of late complications and long-term follow-up.

Esophageal dysmotility is reported in 80–100% of patients and may be the cause of complications and bothersome symptoms [4,5]. Therefore, it is one of the most interesting aspects to study during childhood development. Dysmotility of the esophagus determines various consequences involving the gastrointestinal and the respiratory systems and generally compromises the state of nutrition and growth [6]. As highlighted in a study by Sistonen et al. based on adolescent and adult patients with EA, dysphagia and gastroesophageal reflux were present in 85% and 34% of cases, respectively, with an increased risk of Barrett’s esophagus and metaplasia [7]. These two problems have been related to dysmotility, often characterized by aperistalsis or pan-esophageal pressurization, which make the peristaltic waves ineffective and unable to obtain proper swallowing and esophageal emptying [8,9]. 

Esophageal dysmotility can be related to neuronal abnormalities that occurred during development, post-surgical alterations, and inflammation due to gastroesophageal reflux disease. Its occurrence is probably determined by a set of or “multiple” factors [10]. 

The endoluminal functional lumen imagine probe (EndoFlip™ Medtronic, Crospon Inc., Minneapolis, USA) system is a recent innovative diagnostic tool used to study esophageal function [11,12,13,14,15,16]. Its impedance planimetry system provides real-time, objective feedback on esophageal geometry, evaluating the esophageal luminal cross-sectional area in relation to pressure during controlled volumetric stretches. This phenomenon is also referred to as luminal distensibility. Protocols for EndoFlip™ system applications in the gastroenterology literature have been defined in adults and in small pediatric series [11]. In pediatric age, Menard-Katcher et al. reported data of patients with eosinophilic esophagitis and Ng K et al. described their experience in the use of EndoFlip™ in 18 patients with various esophageal diseases [12,13]. These authors concluded that the method is feasible and safe in pediatric ages and that, for eosinophilic esophagitis, a reduction in distensibility is identifiable even at an early stage of the disease. More recently, an intraoperative use has also been proposed to assess the tightness of a fundoplication or the adequacy of a myotomy [14].

The aim of our study was to report the use of EndoFlip™ in the follow-up of a series of EA patients and to describe the esophageal distensibility patterns, together with motility features of the esophagus. A secondary aim was to determine whether this instrument may be helpful in the functional evaluation of the esophagus in EA patients.

## 2. Materials and Methods

Endoscopic studies of the upper gastrointestinal tract, performed with EndoFlip™ in patients affected by EA with a minimum follow-up of 9 years, were retrospectively evaluated. Data were collected from January to June 2022 at V. Buzzi Children’s Hospital, Milano, Italy. The following data were collected from medical records: demographics, type of EA, associated anomalies, type of primary surgery and/or definitive repair, complications, endoscopic surveillance details, symptoms, perinatal data and auxological data (weight, height, BMI, and BMI score) at the endoscopic evaluation. Follow-up information included the assessment of growth and nutritional status (weight and height), respiratory function (spirometry) and upper gastrointestinal evaluation, as per European guidelines [1]. To describe esophagogastric junction (EGJ) distensibility, we used the reference values reported by Pandolfino et al., who developed distensibility ranges of the EGJ at filling values of 60 mL in healthy adults (Table 1) [11].

We retrospectively analyzed data according to the Declaration of Helsinki, 2008 revision. Sensible data and collected information are blinded, cannot be traced back and they are reserved (General Authorization to Process Personal Data for Scientific Research Purposes-Authorization no. 9/2014; Regulation (EU)/2016/679 GDPR, Legislative Decree n.101/18).

### 2.1. EndoFlip™ Protocol 

We adapted the EF-322 Protocol, suggested by the Medtronic manufacturer and validated by Pandolfino et al. [11]. The procedures were performed under general anesthesia (propofol and midazolam) in spontaneous breathing in all patients but one, for whom the need of an operative endoscopy for esophageal stenosis was expected. We used an EF-322 catheter, consisting of a 16 cm long balloon with 16 impedance sensors spaced 1 cm apart. Catheter calibration was performed via the FLIP Analytics software (Crospon, Inc. Galway, Ireland) prior to the study. The placement of the catheter was preceded by an endoscopic evaluation to accurately detect the position of the Z line and the EGJ. The balloon was placed orally, leaving at least two pairs of impedance sensors distal to the junction, into the stomach. At the desired length, the EF-322 balloon was filled up to 20 mL with 0.3% saline. We waited 15–20 s for stabilization and landmark confirmation. Refilling was then performed using a gradual approach and monitored for 30 s at each fill level: 30 mL, 40 mL, and 50 mL. Once all data were recorded, the balloon was deflated to volumes of 10 mL or less and the catheter was withdrawn. Balloon inflation was stopped earlier if balloon pressure exceeded 60 mmHg, as suggested by the manufacturer’s guidelines. At the end, the endoscopic evaluation with multiple biopsies was completed. We recorded luminal parameters including diameter, compliance, cross-sectional area (CSA), pressure, and distensibility index. Contractile patterns were also evaluated to identify anterograde and retrograde repetitive contractions and an absent or reduced contractile response. To evaluate the esophageal body, we arbitrarily chose to consider the distensibility values measured at the levels of electrode 1, 3, and 5. These three electrodes were far enough from the EGJ so as not to be affected by it. Selected patients with severe dysmotility patterns underwent high-resolution manometry (HRM) study for further assessment of their situation.

### 2.2. Statistical Analysis

Qualitative variables were described as counts and percentages. Quantitative variables were expressed as the mean and standard deviation (SD) or the median and IQR (range) as appropriate. The Shapiro–Wilk’s test was used to test the normality of the data. Statistical analyses were performed using the exact Fisher test and a *p*-value below 0.05 was considered statistically significant. The data analysis was performed with the STATA statistical package (release 15.1, 2017, Stata Corporation, College Station, TX, USA).

## 3. Results

### 3.1. Clinical Data and Surgical History

Table 2 summarizes the demographic, anamnestic, clinical, and surgical features of the nine patients (M:F = 7:2) included in the study. At the time of evaluation, the patients had a median age of 13 years (range 9–18 years). Five patients (55.56%) were born premature (gestational age <37 weeks). According to Gross’s classification, the diagnosis was type A EA in two patients (22.22%), type C EA in 6 cases (66.67%), and type D EA in one case (11.11%). In five cases (55.56%), a primary anastomosis had been possible approximately 24 h after birth, while in the remaining four cases (44.44%) a long-gap atresia had been corrected either by gastric pull-up (3/4 patients) at a median age of 12 months (range 6–12 months), or by Sharli esophagoplasty (1/4 cases), at the age of 3 months. At follow-up, seven patients (77.78%) required a fundoplication for gastroesophageal reflux disease, while five patients (55.56%) developed an anastomotic stenosis that required one or multiple mechanical dilations (median 1; range 0–20) in the first 3 years of life. Seven children experienced at least one episode of bolus impact and reported failure of food transit in the esophagus. In six cases (66.67%), respiratory complications occurred, especially in the first years of life. Patients mainly reported recurrent infections, bronchiolitis, and asthmatic bronchitis.

At the time of the current evaluation, five patients (55.56%) were asymptomatic and four (44.44%) reported sporadic episodes of gastroesophageal reflux and well-compensated dysphagia. Dysphagia was described as difficulties in eating solid and “stringy” foods. (such as meat). 

### 3.2. EndoFlip™ and HRM Data

Table 3 summarizes the main data obtained during the EndoFlip™ evaluation.

At filling values of 40 and 50 mL, the intraluminal pressure (mmHg) reached values greater than 15 mmHg in 88.89% of cases. The median minimum distensibility (calculated at the point with the smallest diameter) was 2.58 mm2/mmHg (range 0.51–4.30). It was observed that the point reported by EndoFlip™ as the least distensible corresponded to EGJ in two cases (22.22%), to anti-reflux plastic level in six cases (66.67%), and to the site of the anastomosis in the remaining case (11.11%). The median distensibility (mm^2^/mmHg) of the esophageal body at electrodes one, three and five were 9.5, 9.44, and 7.9, respectively. At the level of known anastomotic stenosis (radiologically reported in four patients), the median diameter at maximum filling was 13.9 mm (range 13–17.7) with a median distensibility of 6.99 mm^2^/mmHg (range 3.18–28.8). Signs of esophageal peristalsis were present in three patients (33.33%), with a median of 6.5 waves (range 6–7) recorded in a 20 s frame. In the remaining six cases (66.66%) it was not possible to evoke any peristaltic wave, even at maximum filling. In one patient, we found a narrow and minimally distensible EGJ (minimum diameter 7.1 mm and minimum distensibility 1.44 mm^2^/mmHg), with an endoscopic picture of esophagitis and absent peristalsis. The aforementioned patient was a 14-year-old boy, treated at another center, who had undergone a Toupet fundoplication. Table 4 summarizes the HRM parameters in three patients. 

## 4. Discussion

Current literature emphasizes the relevance of long-term follow-up of patients with EA, especially at the age of transition, which is a delicate phase when the body undergoes many changes and functional alterations in which anomalies in development could lead to life-long severe complications [1,4,5,6,17,18,19,20,21]. 

We considered data recorded during planned endoscopic evaluations with EndoFlip™ device in nine patients. In agreement with literature data, we had a higher prevalence of type C EA, five patients with a history of anastomotic stenosis (60% of them following long-gap repair), and long-term complication rates similar to those previously reported [1]. 

We recorded EGJ distensibility median values of 2.58 mm^2^/mmHg, corresponding to the minimum value recorded in 88.89% of cases. These data were found both in patients with and without an anti-reflux plastic, confirming the radiological evidence of good EGJ continence in our population.

Specifically evaluating patients after fundoplication (n = 7), we recorded a median distensibility of 2.12 mm^2^/mmHg (range 1.44–11.1), slightly higher than those described by DeHaan et al. However, DeHaan considered the intraoperative data of adult patients undergoing Nissen (45 patients) and Toupet (30 patients) [22]. Our distensibility data appear to be more similar to those of a normal condition (plastic-free EGJ as described by Pandolfino et al. [11]) that would allow the junction to be continent without inhibiting the passage of food and secretions. Considering the mean diameters, we found similar values to those of DeHaan (9.6 ± 2.0 mm DeHaan vs. 9.21 ± 2.93 mm in our study). However, we should consider the age difference of the two populations: in our series, the EGJ with the same diameter seems to have better compliance (distensibility 2.2 ± 1.3 mm^2^/mmHg DeHaan vs. 3.59 ± 3.21 mm^2^/mmHg in our study) (Table 5) [22].

We found one patient with esophagitis and absent peristalsis and a narrow and not very distensible EGJ. These data would confirm the presence of a partial stenosis of the antireflux valve and the need for surgical intervention, considering partial valves in selected cases [23,24,25,26,27,28,29]. According to the ESPGHAN-NASPGHAN Guidelines, fundoplication is indicated in all EA patients with PPI non-responder reflux, recurrent anastomotic stenosis or cyanotic crises [1]. 

Omari et al. reported the data of manometric studies on 13 pediatric patients undergoing fundoplication with respect to the effects on motility and found impairments in the post-operative period, especially in those with pre-surgical motility issues [25]. For this reason, they recommended performing pre- and post-operative functional assessments to define outcomes and to select surgical indications.

According to Loots et al., in the absence of preoperative dysphagia, hemifundoplication does not cause negative effects on esophageal motility [25].

In light of these data, it is important to evaluate the dysmotility patterns in patients with EA to avoid post-operative worsening of dysphagia after an excessive increase in EGJ pressures with consequent intra-esophageal stagnation of food, esophagitis, and esophageal dilatation.

Furthermore, to prevent this situation, a possible future application of EndoFlip™ could be the pre- and intra-operative study during antireflux surgery. Preoperative evaluation could include study with EndoFlip™ combined with HRM to identify motility disorders and to define the characteristics of the EGJ. Intraoperatively, the EndoFlip™ could guide the surgical technique, allowing for a “tailored” fundoplication.

The distensibility of the esophageal body was calculated as the mean value of three electrodes chosen arbitrarily far away from the EGJ. Almost all studies in literature focus only on EGJ distensibility; only two papers consider data from the esophageal body in adults with eosinophilic esophagitis [26,27].

Nicodeme et al. and Carlson et al. conclude that in eosinophilic esophagitis patients, there is a greater rigidity of the esophageal wall with an increased risk of bolus impact and the need for dilatation [28,29]. They also found that the distensibility value is influenced by esophageal contractility and respiratory parameters.

In our series, the esophageal body distensibility was higher than that of the EGJ, as expected on the basis of the well-known anatomical notions that describe EGJ as an anatomo-functional sphincter. Esophageal body distensibility data seem to be within a normal range. This supports the innovative theories on the etiology of EA dysmotility, which implicate not only surgical factors (rigid walls due to ischemic problems) but also intrinsic neural alterations (lower density of interstitial cells of Cajal, imbalance in neurotransmitter secretion, abnormal branching of the vagus nerve, and hypoganglionosis) [30,31,32].

Four patients reported mild symptoms at the time of EndoFlip ™ evaluation. 

Two patients presented heartburn and food regurgitation after large meals: one of them had valid peristaltic waves with continent and distensible (3.67 mm^2^/mmHg) EGJ, while the other had distensible EGJ (2.69 mm^2^/mmHg after Nissen fundoplication) with no peristalsis, indicating a possible correlation between symptoms and dysmotility.

Two patients had dysphagia: one with a very reduced EGJ distensibility (0.51 mm^2^/mmHg), attributable to the inflammation and edema (LA2B esophagitis) found during endoscopy and the other with esophageal dysmotility and apparently preserved EGJ distensibility. Indeed, the esophageal body can appear as a constricted and non-distensible conduit or as an ectatic channel. The latter has initially high distensibility values without a proper response. Moreover, in patients who had extensive gastric mobilization, the distensibility dynamic may refer to a “new esophagus” that is the stomach itself with specific distensibility patterns. 

The presence of anastomotic stenosis was radiologically identified in four patients. The stenotic segments were easily distensible during EndoFlip™ (mean 5.72 ± 1.89 mm^2^/mmHg), confirming the clinical absence of symptoms. However, dysmotility remains in these fibrotic areas, with the risk of affecting adjacent tracts.

In two patients, we noticed that there was no correlation between the distensibility results and the severity of the stenosis on radiological evaluations. Unfortunately, there are no reported data on this aspect. The EndoFlip™ results seem to correlate better with the clinical picture and would help to define the management program and the possible efficacy of dilatations.

We acknowledge some limitations of our study, mainly related to the small size of the sample and the lack of reference data for healthy pediatric subjects. The small size of the sample is due to the rarity of the disease (1/2500 live births according to national data). Moreover, the lack of centralization in our country involves the management of patients by several pediatric surgery centers, further reducing the annual number of referred patients. 

Furthermore, during data analysis we were unable to normalize the values based on esophageal peristalsis and respiration, although all the procedures were performed under the same sedation conditions. This may be partially explained by individual differences in respiratory parameters, but it was mainly a decision made by the anesthesiologist for safety reasons. 

This work represents a preliminary evaluation that can be the beginning of future multicenter studies. The inclusion of a control group would highlight the peculiar features of the EA patient, helping to build tailored approaches, beginning in the neonatal period. Comparative analysis with HRM is extremely important for clarifying the functional aspects and providing specific clinical implications. 

## 5. Conclusions

We have studied the applicability of a relatively new instrument to evaluate esophageal distensibility during the follow-up of EA patients. EndoFlip™ was both safe for and applicable to children. It allowed us to measure the esophageal distensibility and diameter to indirectly deduce some characteristics of motility in terms of peristalsis.

We found that the distensibility of the esophageal wall is within the normal range in most cases, both at the level of the gastroesophageal junction and of the body, with clinical implications. In case of stenosis, EndoFlip™ associated with other diagnostic methods (digestive tract X-ray, EGDS, and HRM), was helpful in defining the management and timing of future evaluations.

Some very interesting data relate to the study of EGJ intraoperatively during antireflux surgery and follow-up. 

In the future, EndoFlip™ could be routinely used during the follow-up of patients with AE, although considerable further research should be carried out to understand esophageal dysmotility in EA patient outcomes.

## Figures and Tables

**Table 1 children-09-01426-t001:** EndoFlip™ distensibility ranges at the esophagogastric junction [11].

Distensibility (mm^2^/mmHg)	
Reduced	<2
Borderline	2.1–3.0
Normal	3.1–9.0
Increased	>9.1

**Table 2 children-09-01426-t002:** Clinical data of esophageal atresia patients considered in the current series.

Pt	Sex	Age (Years)	EA Type	Long-Gap	Surgery	Associated Malformations	Fundoplicatio (Age)	Stenosis	BMI (Z-Score)	Symptoms
1	M	14	C	No	Primary anastomosis	Cardiologic	Toupet (1 year)	Yes	19.13 (0.03)	Dysphagia
2	F	15	C	No	Primary anastomosis	Cardiologic	Nissen (15 years)	No	19.67 (−0.1)	-
3	M	15	C	Yes	Gastric pull-up	Cardiologic Renal	Nissen (1 year)	Yes	19.97 (0.63)	-
4	M	10	C	No	Primary anastomosis	Microcephalia	Nissen (1 year)	Yes	16.84 (0.55)	-
5	M	9	C	No	Primary anastomosis	ARM, tethered cord	-	No	14.2 (−1.09)	Dysphagia
6	M	13	D	No	Primary anastomosis	Intestinal malrotation, duodenal stenosis	-	No	18.89 (0.12)	Pirosis, regurgitation
7	M	12	A	Yes	Gastric pull-up	Subclavia lusoria	Nissen (1 year)	Yes	16.02 (−0.95)	Pirosis, regurgitation
8	M	17	C	Yes	Gastric pull-up	Duodenal atresia, scoliosis	Nissen (2 years)	No	15.54 (−2.95)	-
9	F	21	A	Yes	Sharli esophagoplasty	Cardiologic, scoliosis	Dor (1 year)	Yes	21.64 (−1.74)	-

**Table 3 children-09-01426-t003:** Endoflip™ data. Distensibility parameters, diameters and peristalsis compared to the clinical picture.

Pt	Filling (mL)	Pressure (mmHg)	Min Diameter	Min Distensibility	Anatomical Landmark	EB Distensibility			Peristalsis	Num of Waves in 20 s	Stenosis	Symptoms	Food Impaction	Dilatations
						Electrode 1	Electrode 3	Electrode 5						
1	50	27.5	7.1	1.44	Toupet	9.5	7.72	5.97	Absent	-	Yes	Dysphagia	-	1
2	50	42.5	9.2	1.56	Nissen	3.73	4.67	5.28	Yes	6	-	No	Si	-
3	40	17.2	13/15.60	7.72/11.11	anastomosis/Nissen	11.4	7.72	9.87	Yes	-	Yes	No	Si	2
4	50	32.3	10.3	2.58	Nissen	6.3	9.44	8.14	Absent	-	-	No	Si	6
5	40	33.8	4.7	0.51	EGJ	6.17	10.44	7.2	Absent	-	Yes	Dysphagia	Si	-
6	50	29.8	11.8	3.67	EGJ	5.77	6.83	5.77	Yes	7	-	Pirosis. regurgitation	-	-
7	50	24.4	9.1	2.69	Nissen	13.11	11.11	7.9	Absent	-	-	Pirosis. regurgitation	Si	20
8	50	8.7	6.9	4.3	Nissen	35.75	37.57	29.9	Absent	-	Yes	No	Si	-
9	40	21.5	6.3	1.45	Dor	10.19	19.16	17.84	Absent	-	-	No	Si	4
Median [range]		27.5 [8.7–42.5]	9.1 [4.7–13/4.7–15.6]	2.58 [0.51–7.72/0.51–11.11]		9.5 [3.73–35.75]	9.44 [4.67–37.57]	7.9 [5.28–29.9]						


**Table 4 children-09-01426-t004:** HRM findings in three patients of the reported series.

		Patient 2	Patient 3	Patient 8
LES lenght (cm)		2.6	3.1	2.4
Intra-abdominal LES (cm)		0	0.8	2.1
IRP of basal LES		46	16.9	54.7
IRP of basal UES		98.2	39.7	
Median IRP		41.3	1.9	57.2
Chicago Classification				
	failed	33%	50%	67%
	weak	0	12%	0
	ineffective	33%	62%	67%
	panesophageal pressurization	33%	0	67%
	premature contraction	33%	12%	33%
	rapid contraction	33%	12%	22%
	fragmented	0	0	0
	intact	67%	38%	33%
DL		3.3	5.3	−1.6
Peristalsis on EndoFlip^TM^		Yes	Yes	No

**Table 5 children-09-01426-t005:** Diameter and distensibility parameters (mm^2^/mmHg) in the current study compared to those found by DeHaan et al. [22].

		Diameter	Distensibility
Post-fundoplication (DeHaan)	Total (n = 75)	9.6 ± 2.0	2.2 ± 1.3
	Nissen (n = 45)	9.1 ± 1.8	2.0 ± 1.4
	Toupet (n = 30)	10.6 ± 2.1	2.4 ± 1.1
Current study	Total (n = 7)	9.21 ± 2.93	3.59 ± 3.21

## Data Availability

Data supporting reported results are archived in the first author’s personal datasets.

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
