# Peer review of "Utility of Functional Lumen Imaging Probe in Long-Term Follow-Up of Children with Esophageal Atresia: A Single-Center Retrospective Study"

_children, 2022, doi:10.3390/children9101426_

Round 1
Reviewer 1 Report
First and foremost, I'd like to congratulate the authors on such an intriguing study. Even though the EndoFlip device has been available for several years, it has not been widely used in the pediatric population.
This is one of the reasons why the article may be accepted for publication. Otherwise, as the authors noted in the limitations section of their report, the number of patients is small and the results are not reproducible.
It is a good starting point for larger, possibly multicentric studies. The authors' work should be expanded, and one suggestion is to compare the results with a control group. Also, consider conducting a comparative analysis with HRM, for example.
Author Response
First and foremost, I'd like to congratulate the authors on such an intriguing study. Even though the EndoFlip device has been available for several years, it has not been widely used in the pediatric population. This is one of the reasons why the article may be accepted for publication. Otherwise, as the authors noted in the limitations section of their report, the number of patients is small and the results are not reproducible.
It is a good starting point for larger, possibly multicentric studies. The authors' work should be expanded, and one suggestion is to compare the results with a control group. Also, consider conducting a comparative analysis with HRM, for example.
Thank you. We agree with you in believing that the inclusion of more patients, considering multicentric projects, would bring further implications. The inclusion of a control group would highlight the peculiar features of the EA patient helping in building tailored approaches, starting from the neonatal period. Comparative analysis with HRM is extremely important to clarify the functional aspects and brings specific clinical implications. We’ve added these considerations in the manuscript. See pag. 8, line 264 and following.
Reviewer 2 Report
Congratulations to the authors on this exciting study. Destro et al. evaluated the utility of functional lumen imaging probes in long-term follow-up of children with esophageal atresia. Overall, the theme is relevant. However, I have some remarks.
There are numerous typos in the manuscript, and English should be reviewed.
In the Abstract, the authors report that they included only patients with at least nine years of follow-up. This information is not found in the body of the manuscript.
In the Results, the follow-up of each patient should be presented.
Instead of presenting the mean and its corresponding standard deviation, I suggest using median and range intervals since only nine patients were included in the study,
In the Discussion section, the study’s limitations should be better discussed.
The conclusions are too long. The authors should focus on answering their initial objectives.
Author Response
Congratulations to the authors on this exciting study. Destro et al. evaluated the utility of functional lumen imaging probes in long-term follow-up of children with esophageal atresia. Overall, the theme is relevant. However, I have some remarks.
Thank you very much for your comment.
There are numerous typos in the manuscript, and English should be reviewed.
The manuscript was reviewed by a native speaker, who was acknowledged at the bottom of the manuscript.
In the Abstract, the authors report that they included only patients with at least nine years of follow-up. This information is not found in the body of the manuscript.
The information was added in the manuscript, as well. (page 2, line 79)
In the Results, the follow-up of each patient should be presented.
Follow-up data are listed in the Results starting from page 4, line 141 and resumed in Tab. 2
Instead of presenting the mean and its corresponding standard deviation, I suggest using median and range intervals since only nine patients were included in the study,
Median and range intervals were used throughout the manuscript instead of mean and sd.
In the Discussion section, the study’s limitations should be better discussed.
Limitation session was expanded, considering also the comment of rev1 (see page 8 line 264 and following)
The conclusions are too long. The authors should focus on answering their initial objectives.
Conclusions were reduced focusing on study objectives. (page 8, lines 281-294)